# Analysis of Factors Affecting Termite Damage to Wooden Architectural Heritage Buildings in Korea

**Si-Hyun Kim** [1] and **Yong-Jae Chung** [2,*]

1 Safety and Disaster Prevention Division, National Research Institute of Cultural Heritage, Daejeon 34122, Korea; shkim1242@korea.kr
2 Department of Heritage Conservation and Restoration, Graduate School of Cultural Heritage, Korea National University of Cultural Heritage, Buyeo-gun 33115, Korea
* Correspondence: iamchung@nuch.ac.kr

**Abstract:** Many wooden architectural heritage buildings exist in Korea, and the authenticity and structural stability of these cultural assets are being affected by termites. This study aimed to identify the degree of termite damage and related factors in these buildings. The degree of termite damage to 182 nationally designated wooden architectural heritage buildings (national treasures and treasures) in Korea was quantified, and data were collected for 11 factors affecting termite damage, such as the surrounding environment and architectural features. Based on the results of a general linear model analysis, the following three factors were identified to have a significant effect on termite damage: the type of contact between the ground and wooden structural items, the number of days of termite activity, and the proportion of forests in the land surrounding the property. This study is the first attempt to statistically analyze factors affecting termite damage to wooden heritage buildings, and our results provide initial data for the preservation and management of these properties.

**Keywords:** termite; wooden architectural heritage; subterranean termite; *Reticulitermes speratus kyushuensis*

## 1. Introduction

Wood is a readily available natural material that is used for many purposes. In East Asian countries, including South Korea, wood and soil were once the main building materials, and a number of wooden heritage buildings remain [1].

Wooden architectural heritage buildings are subjected to damage and declining authenticity due to various factors. Among these, wood-rotting fungi and termites are the most important [2]. Termites are a group of social insects that use intestinal symbionts or enzymes to digest various plant materials. Although termites play an important role as decomposers in a forest ecosystem, they inflict economic damage by harming wooden buildings and crops [3]. The global economic loss due to termites amounts to USD 40 billion annually, 80% of which is caused by subterranean termites [4].

The damage caused by termites not only represents an economic loss but also affects many wooden buildings associated with cultural heritage. In particular, the damage is severe in East Asia and Southeast Asia, where many wooden architectural cultural assets remain and termites actively breed in the warm climate [5]. In Japan, a survey by the Agency for Cultural Affairs between 1971 and 1973 revealed damage to 42.6% of approximately 2000 cultural properties, and termite damage to many other wooden cultural buildings has also been identified [6]. Termite damage to cultural properties is common in China, particularly in the southern regions [7], and termite damage has also been reported for many wooden cultural buildings in Taiwan [8].

Three species of termites are distributed within Korea. In the early 1900s, *Reticulitermes speratus kyushuensis*, a type of subterranean termite, was identified nationwide [9,10], whereas the species *Reticulitermes kanmonensis* and *Glyptotermes nakajimai* were more recently

discovered [11,12]. Termite damage has been confirmed in wooden cultural buildings in Korea since the 1970s. In a 2009 study, termite detection was confirmed by dog responses in 78 of 231 buildings [13]. In a survey conducted by the National Research Institute of Cultural Heritage (NRICH) from 2016 to 2019, termite damage was confirmed by termite detection dog responses in 317 of 362 (87.6%) cultural properties, and there were 185 cases (51.1%) of visible termite damage [14].

The original form and structural stability of wooden cultural buildings may become damaged by termites. As such damage tends to be irreversible, it is important to prevent it. Therefore, to establish effective termite control measures for nationally designated architectural heritage properties in Korea, this study aimed to identify the factors affecting termite damage by statistically analyzing architectural characteristics and environmental factors.

## 2. Materials and Methods

### 2.1. Subjects

Numerous wooden cultural buildings remain in Korea, and among them, those of a particular historical value have been categorized and managed as nationally designated architectural heritage properties by the Cultural Heritage Administration of Korea (Figure 1). This study included 182 properties designated by the Korean government, 25 national treasures (heritage of a rare and significant value in terms of human culture and with an equivalent value to Treasure) and 157 treasures.

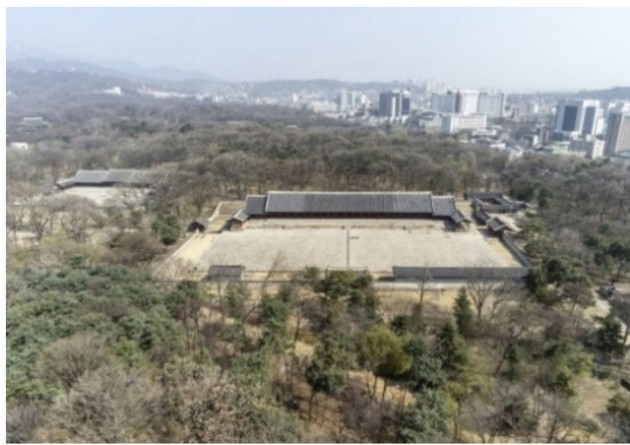 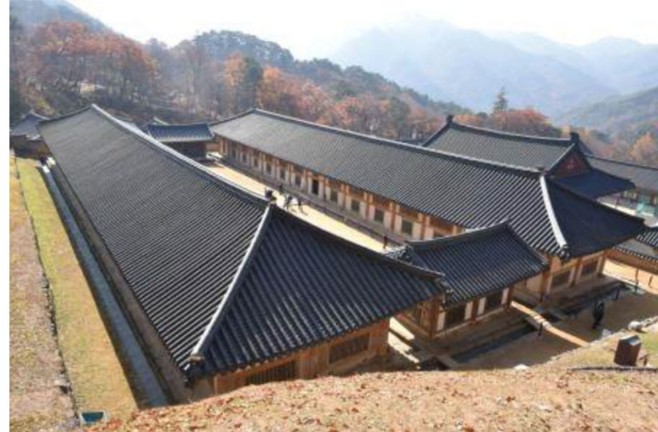

**Figure 1.** Wooden architectural heritage buildings in Korea. (**Left**) Jongmyo Royal Shrine (National Treasure of Korea, UNESCO world Heritage). (**Right**) Janggyeong Panjeon depository of the Tripitaka Koreana (National Treasure of Korea, UNESCO world heritage).

### 2.2. Quantification of Termite Damage to Architectural Heritage Properties

In the "Report on Biological Damage to Wooden Cultural Properties", which related to a study conducted by the National Research Institute of Cultural Heritage from 2016 to 2019, the degree of termite damage for each cultural property was identified using human eyesight and termite detection dogs [15–29]. In this report, the subjects of the investigation were pillars and Hainbang(the horizontal member between lower parts of pillars), which are mainly damaged by subterranean termite. The degree of termite damage for each cultural property was calculated as the sum of the number of structural members for which termite damage was confirmed with human eyesight and the number of structural members whose damage was confirmed by termite detection dogs, divided by the total number of structural members.

### 2.3. Data Collection Details

This study selected 11 factors that were expected to affect the distribution and damage caused by termites: the number of days of termite activity, the equilibrium moisture content

(EMC) of wood, the distance of the architectural heritage property from forests, the area of forests surrounding the architectural heritage property, the distance from water systems, the type of architectural heritage property, the time elapsed since full repair or restoration, the type of contact between the ground and wooden items, the distance between the ground and wooden items, heating arrangements, and occupancy. Related data for these factors were collected for each wooden architectural heritage property.

### 2.3.1. Number of Days of Termite Activity

Temperature directly affects the range or duration of termite activity [30,31]. Based on the information provided in previous studies, the average daily temperature for each architectural heritage property was estimated using location and meteorological data, and the number of days during which termites were potentially active was determined [32]. First, the location (latitude, longitude, and elevation) and temperature data of all meteorological stations with daily average temperature data for 10 years (2006–2015) were obtained from the open meteorological data portal of the Korea Meteorological Administration (https://data.kma.go.kr/ (accessed on 23 August 2021)). A total of 131 meteorological stations were selected, excluding those with locations that had been moved more than 10 m or with daily average temperature data that were missing for more than 20 days. The daily average temperature from 1 January to 31 December was calculated for 10 years at each weather station. Then, the daily average temperature was used as the dependent variable in a multiple regression analysis, with latitude, longitude, and elevation as independent variables, to obtain the multiple regression equation for estimating the daily average temperature for each day.

The daily average temperature for each architectural heritage property was estimated by substituting the locational information for each architectural heritage property into the calculated multiple regression equation for each day. The location information of each architectural heritage property was extracted from the "Report on Biological Damage to Wooden Cultural Properties" [15–29]. Using the estimated daily average temperature, the number of days with an average temperature of 6°C or higher for each architectural heritage property was identified. This is considered the minimum suitable temperature for termite activity based on a previous study on *R. s. kyushuensis* [33].

### 2.3.2. Equilibrium Moisture Content of Wood

Subterranean termites prefer a high-humidity environment and wood with a high moisture content to retain moisture in their bodies. Therefore, the moisture content of wooden architectural heritage buildings affects the degree of termite damage. As the dead wood moisture, which is the actual moisture content of wood, converges to the EMC over time, the EMC was used as a measure of moisture content in this study. In a previous study, the EMC values of wood for 73 regions across the country were presented using climate data from 1980 to 2010 [34]. These values were processed in the same way as the temperature to calculate the cumulative EMC for each architectural heritage property from April to October, which is the typical period for termite activity.

### 2.3.3. Distances from Forests and Water Systems, and Areas of Forests Surrounding the Properties

Forests have the highest density and diversity of termites among various landscapes [35], and there is a risk that termites from forests will migrate to heritage properties; many of these properties are located in areas adjacent to forests [36]. In this study, information regarding how far each architectural heritage property was from adjacent forests and water systems, and the proportion of forests in the land surrounding each architectural heritage property, was analyzed using the geographic information system (GIS) program ArcGIS 10.3.

Each building figure was created using location information on each architectural heritage property, which was overlaid with the land cover map obtained from the Environ-

mental Geospatial Information Service (https://egis.me.go.kr/ (accessed on 14 September 2021)) of the Ministry of Environment for analysis. The 2019 "Sub-divided Land Cover Map" was used to extract data on those broadleaf and coniferous forests with a major classification on the land cover map of "forest" and the shortest distance between them and each architectural heritage property was determined.

For information regarding the distance to a water system, the shortest distance between data points on rivers and lakes (where the major classification on the land cover map was "water") and architectural heritage properties was extracted. The area of forest surrounding cultural properties was calculated by extracting all land cover maps within a radius of 1 km of each architectural heritage property and designatingthe land coverasurban areas, farmland, forest, grassland, wetland, bare land, and water systems.

### 2.3.4. Type of Architectural Heritage Property

Architectural heritage properties have different architectural characteristics depending on their original function, which may affect the damage caused by termites. According to the classification criteria of the electronic administrative portal of the Cultural Heritage Administration, architectural heritage properties are classified into seven types for analysis: houses, palaces and government offices, pavilions and gazebos, graves and shrines, temples, schools and lecture halls and fortress walls.

### 2.3.5. Time Elapsed since Full Repair or Restoration

Depending on the extent of damage to the constituent materials, wooden architectural heritage properties maybe dismantled and repaired once or more over a period of tens to hundreds of years. Because termite damage accumulates over time, this study also confirmed when the buildings, including their columns, were last repaired through the periodic survey of individual cultural properties, precise measurement reports, repair reports, and applications for designation as national cultural properties. As there were many cultural properties that did not have a record of restoration before the 1990s, this factor was classified into three groups: 0–20 years, 20–40 years, and >40 years since the last dismantling and repair.

### 2.3.6. Type of Contact between the Ground and Wooden Items

The termites that cause damage to wooden cultural properties in Korea are subterranean. Due to the ecological characteristics of subterranean termites, they are introduced into cultural properties through the ground under the building during their foraging after establishing a habitat near or under the building. Therefore, the ways in which the wooden items in the lower section of a building are in contact with the ground can affect the risk of termite infestation. Therefore, in this study, each architectural heritage property was classified to one of three types by analyzing the contact method between the ground and the lower sections of the wooden structures based on periodic surveys and detailed survey reports (Figure 2), as follows:

1.  Wooden items are in direct contact with the ground (left);
2.  Wooden items are connected to the ground via materials termites can pass through, such as clay in wood-burning stoves or furnaces (center);
3.  Wooden items are connected to the ground via materials such as stone or brick, which termites cannot pass through, or are totally separated from the ground (right).

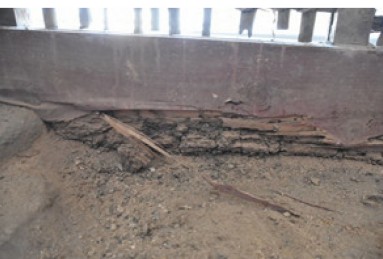 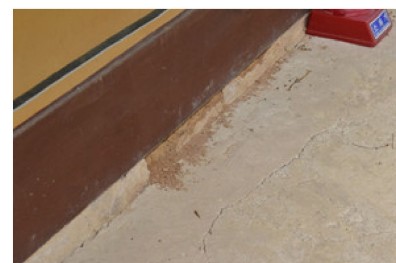 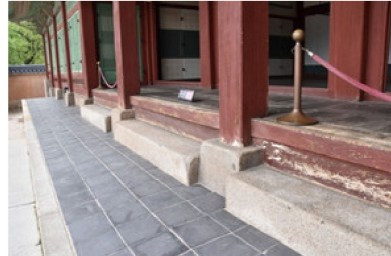

**Figure 2.** Types of contact between the ground and wooden items in wooden cultural buildings in Korea.

### 2.3.7. Distance between the Ground and Wooden Items

When subterranean termites enter a wooden building, a greater distance between the ground and wooden items in the building (the podium or lower columns) increases their travel distance, reducing the risk of infestation. Therefore, in this study, the distance from the ground to the lowest part of the wooden item closest to the ground was obtained and used for analysis through precise measurement reports and actual measurement drawings of individual architectural heritage properties. The total distances from the ground to the bottom of the lowest wooden item were classified into four groups: 0–224 mm (63 buildings), 224–406 mm (63 buildings), 415–4300 mm (63 buildings), and >4300 mm (7 buildings). The last 7 buildings have wooden structures on top of the castle walls.

### 2.3.8. Heating and Occupancy

Some wooden cultural buildings, such as houses, are currently inhabited and therefore equipped with heating devices. Various other features for inhabitant comfort and convenience may also be installed, and condition changes occur through daily management activities inside and outside of the buildings. Therefore, the temperature and relative humidity inside the wooden building may be maintained at levels more suitable for termite habitation that those in uninhabited buildings. Thus, this study aimed to determine if human occupancy or heating in architectural heritage properties affects termite damage. The occupancy and heating status of each architectural heritage property included in the "Report on Biological Damage to Wooden Cultural Properties" (NRICH, 2017; 2018; 2019; 2020) were used in this analysis.

### 2.4. Statistical Analysis

Using the 11 types of collected data as independent variables and the degree of termite damage to individual architectural heritage properties as the dependent variable, statistical analysis was conducted using a general linear model (GLM). The final model was obtained by performing model fitting, including removing variables with low statistical significance using forward stepwise selection. The statistical software used was SPSS 21(IBM, Armonk, NY, USA).

## 3. Results and Discussion

### 3.1. Statistical Analysis Results

Among the 11 factors analyzed, the three with the most statistically significant effects based on the results of the GLM analysis were the type of contact between the ground and wooden items, the number of days of termite activity, and the proportion of forests in the land surrounding the architectural heritage property (Table 1). In the overall model, $\eta^2$, which represents the effect size of each factor, was 13% for the ground–wooden item contact, 6.5% for the number of days of termite activity, and 3.5% for the proportion of surrounding forests. In general, an effect sizeof 14% or more is considered to have high explanatory power, 6%–14% is considered intermediate, and under 6% is considered marginal [37]. In this model, the ground–wooden item contact and number of days of termite activity had

an intermediate explanatory power, and the proportion of surrounding forests showed a marginal explanatory power.

**Table 1.** Results of the general linear model for ground–wooden item contact, number of days of termite activity, and proportion of surrounding forests as independent variables, and the degree of termite damage as the dependent variable ($R^2 = 0.245$, revised $R^2 = 0.229$).

| Independent Variable | F | p | $\eta^2$ |
|---|---|---|---|
| Ground–wooden item contact | 14.258 | 0.000 | 0.130 |
| Number of days of termite activity | 13.277 | 0.000 | 0.065 |
| Proportion of surrounding forest | 7.018 | 0.009 | 0.035 |
| Constant | 6.159 | 0.014 | 0.031 |

The results of this analysis suggested that the most important factor in the model was the type of contact between the ground and wooden items. The degree of termite damage was the highest for cases in which wooden items were in direct contact with the ground, followed by cases in which wooden items were connected to the ground through materials such as clay in wood-burning stoves or furnaces. The least termite damage was observed in cases in which the wooden items were completely separated from the ground or blocked by materials such as stones or bricks that termites could not pass through. As subterranean termites move underground and enter a building through the base of the lower part of the wooden building, the lower columns and the podium close to the ground in the wooden building were most likely to be damaged [5,38–40]. This implies that materials termites cannot pass through, such as stone pillars or tympanum stone, are present between the ground and wooden items. Alternatively, if the ground and wooden items are completely separated, as in the lower floors of a pavilion, it is difficult for termites underground to access the wooden item on the ground. These results suggest that the damage caused by subterranean termites can be reduced by separating the ground from wooden items using a physical barrier or termiticide treatment on the ground prior to constructing buildings.

The second factor of the model, the number of days of termite activity, had a statistically significant effect on the termite damage estimation model. However, the effect size was lower than that of the contact between the ground and wooden items. Temperature affects the internal enzymes or symbionts of termites, thereby affecting the activity and distribution of each termite species [30,41]. It is also associated with overall termite ecology, including their internal metabolism [31,42,43], the amount of food they eat [33], foraging activities [44,45], and the amount of spawning in outdoor colonies [46]. Therefore, it seemed reasonable for temperatures to have a significant effect on the degree of termite damage. The small effect size may have been due to the narrow range in the number of days of termite activities for surveyed wooden cultural buildings—from a minimum of 223 days to a maximum of 305 days.

The third factor that affected the degree of damage caused by termites—albeit marginally—was the proportion of forests surrounding the wooden cultural building. Forests provide good conditions for termites to inhabit, with the vegetation lowering wind speed and blocking sunlight to maintain high humidity [47]. These conditions also harbor an abundance of food for termites, such as dead trees, fallen leaves, and soil organic matter. Accordingly, the diversity and density of termites in forests are higher than those in other places, possibly leading to termite colonies gradually spreading to the surrounding areas [35,48]. Termites originating in surrounding forests may travel to and damage nearby wooden cultural buildings [36]. This suggests that new termite colonies may enter property from the surrounding forest even after existing termite colonies are exterminated. Therefore, termite control in architectural heritage propertiesrequires continuous management, and the scope of control in the termite control plan should include the surrounding forest and other buildings (Area Wide-Integrated Termite Management) [49,50].

*3.2. Equation for Estimating the Degree of Termite Damage to Wooden Cultural Buildings*

According to the previous analysis, three factors that had a statistically significant effect on the degree of termite damage to architectural heritage properties, and their explanatory power, were identified using a general linear model. Based on these factors, a model estimating the degree of termite damage to individual architectural heritage properties was established (Equation (1)).

The degree of termite damage (%) to a specific architectural heritage propertycan be calculated by determining the sum of the following: the number of days of termite activity multiplied by 0.272, the proportion of surrounding forests multiplied by 0.083, and a variable depending on the type of contact between the ground and the wooden items. The value of this variable was −60.993 for wooden items completely separated from the ground, −56.602 for connection via soil, such as a furnace or a wall, and −23.344 for items in direct contact with the ground. The equation for estimating the degree of termite damage in wooden cultural buildings is as follows:

$$\text{Termite damage degree (\%)} = \text{Ground-wood contact type} + (0.272 \times \text{Number of days of termite activity}) + (0.083 \times \text{Proportion of surrounding forest}) \tag{1}$$

**4. Conclusions**

This study aimed to analyze the factors affecting the degree of termite damage in wooden architectural heritage properties of Korea. Among the 11 factors reflecting the surrounding environment or architectural elements related to termite damage, type of contact between the ground and wooden items, number of days of termite activity, and proportion of surrounding forests were found to havestatistically significant effects. Among these, the type of contact between the ground and woodenitemshad the greatest influence ($\eta^2 = 0.130$), whereas the number of days of termite activity ($\eta^2 = 0.065$) and the proportion of surrounding forest ($\eta^2 = 0.035$) had marginal effects. There are other factors (termite density, soil properties around buildings, rainfall, etc.) that may affect termite damage, although they are difficult to measure and were not included in this study. Therefore, it is difficult to completely estimate the total termite damage using the above three factors. The findings of this study can be useful for the future preservation and management of architectural heritage properties. Previous termite control included the application of bait or termiticide on the soil around the building [51]. This method cannot control subterranean termites that have already entered the lower part of the building. Therefore, it is necessary to prevent termites from directly accessing the timber from the soil. To this end, when dismantling and repairing a wooden building, which can take place every 50 to 100 years, it is necessary to install a physical barrier on the ground or apply termiticide to the top of the soil [52]. These methods can prevent the introduction of subterranean termites by separating the soil and wood, thereby contributing to preservation of cultural properties.

**Author Contributions:** Conceptualization, S.-H.K. and, Y.-J.C.; methodology, S.-H.K. and Y.-J.C.; validation, S.-H.K. and Y.-J.C.; formal analysis, S.-H.K.; investigation, S.-H.K.; writing—original draft preparation, S.-H.K. and Y.-J.C.; writing—review and editing, S.-H.K. and Y.-J.C.; visualization, S.-H.K.; funding acquisition, S.-H.K. All authors have read and agreed to the published version of the manuscript.

**Funding:** This study was supported by National Research Institute of Cultural Heritage (NRICH-2205-D12F-1).

**Institutional Review Board Statement:** Not applicable.

**Informed Consent Statement:** Not applicable.

**Data Availability Statement:** Data are available on request from the 1st author.

**Acknowledgments:** The authors are thankful to administrative support by National Research Institute of Cultural Heritage and Korea National University of Cultural Heritage.

**Conflicts of Interest:** The authors declare no conflict of interest.

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
