# Peer review of "Analysis of Factors Affecting Termite Damage to Wooden Architectural Heritage Buildings in Korea"

_forests, doi:10.3390/f13030465_

Round 1
Reviewer 1 Report
The current study conducted by Kim and Chung investigated the factors affecting termite damage to wooden architectural heritage buildings in Korea. Overall, the study is well presented and suitable for publishing in Forests. However, I suggest that authors consider addressing the points given below before the final publication.
It is still unclear how termite damage was quantified in the MM section. Some details should be added as given reference might not be accessible to every reader. Why did the authors not identify the termite species if there were some live termites in or around the structures? Authors have not considered individual components of each building, e.g., Floor type, wall type, etc., that might also contribute to the termite damage or their incidence. Authors have not reported where termite damage was located in buildings, e.g., inside, outside, or both? Moreover, the authors have not considered the age of the structure in their analysis. What about any hidden damage in the structure? Was any termite protection method used when structures were repaired? Further comments are given below
Page 2
- Line 48: Add authority after the name of each termite species
- Line 54 “of termite damage that was visible to the naked eye” replace it with “of visible termite damage
- Line 56-57: please re-phrase
- Line 67, remove “the subject of”
- Line 68, remove “such”
- Line 69, “national treasures” has been defined in the previous line what are “treasures”?
- Line 77-79, I would suggest adding a summary of guidelines used to quantify or investigate the termite’s damage. Given references will not be accessible to every reader
- Line 80, why authors did not consider the age of the structures? probability of termite attack is also a function of structure age
Page 3
- Line 90, remove “most”
- Line 94-95, replace “First, the location and temperature data (latitude, longitude, and elevation) of all meteorological stations” with “First, the location (latitude, longitude, and elevation) and temperature data of all meteorological stations.”
Page 4:
- What about frame type, floor type, wall type in each building, amount of wood, etc.? Were all of the wood? Or masonry or stone etc. I think these factors are important and should be considered or discussed in this study.
Page 9
- Line 379, correct “possibility”
Please double-check all references
Author Response
Thanks for the detailed review. Responses to review comments are in the attached file.

Reviewer 2 Report
„Analysis of Factors Affecting Termite Damage to Wooden Architectural Heritage Buildings in Korea” is an interesting attempt to establish the model for prediction of sustainability of wooden historical buildings to termite attack in order to protect them against destruction. The research is important from the conservation perspective. However, I have some concerns about the basis of the equation provided and the scientific soundness of the conclusions made. More detailed comments and suggestions can be found in the pdf file attached.

Author Response

(The authors gave the same response as above.)

Round 2
Reviewer 2 Report
In the last sentence of the manuscript, the authors claim that “The findings of this study can be useful for the future preservation and management of architectural heritage properties.” Please explain how you plan to use the results obtained in the wood preservation or delete this sentence. In my opinion, the data obtained are not helpful in the conservation/preservation process, just because nobody will deforest the area around historical buildings or change the type of connection between the building and the ground. It is interesting to know which factors are more or less correlated with the damage by termites, but the findings of this study are not scientifically strong enough to be useful in conservation practice.
Author Response
Thanks for the review. In accordance with the reviewers' opinions, the conclusion was reinforced with how the results of this study can contribute to the conservation and management of actual cultural properties.